# Functional Study on the Key Gene *LaLBD37* Related to the Lily Bulblets Formation

**DOI:** 10.3390/ijms25179456

**Published:** 2024-08-30

**Authors:** Xinru Hou, Kewen Zhang, Yingmin Lyu

**Affiliations:** Beijing Key Laboratory of Ornamental Plants Germplasm Innovation & Molecular Breeding, China National Engineering Research Center for Floriculture, College of Landscape Architecture, Beijing Forestry University, Beijing 100083, China

**Keywords:** oriental hybrid lily, bulblets genesis, *LaLBD37*, homologous transformation, functional verification

## Abstract

Oriental hybrid lilies, known for their vibrant colors, diverse flower shapes, and long blooming seasons, require annual bulb propagation in horticultural production. This necessity can lead to higher production costs and limit their use in landscaping. The LA hybrid lily ‘Aladdin’ has shown strong self-reproduction capabilities in optimal cultivation environments, producing numerous high-quality underground stem bulblets. This makes it a valuable model for studying bulblet formation in lilies under natural conditions. Through transcriptome data analysis of different developmental stages of ‘Aladdin’ bulblets, the *LaLBD37* gene, linked to bulblet formation, was identified. Bioinformatics analysis, subcellular localization studies, and transcriptional activation activity tests were conducted to understand the characteristics of *LaLBD37*. By introducing the *LaLBD37* gene into ‘Sorbonne’ aseptic seedlings via Agrobacterium-mediated transformation, resistant plants were obtained. Positive plants were identified through various methods such as GUS activity detection, PCR, and fluorescence quantitative PCR. Phenotypic changes in positive plants were observed, and various physiological indicators were measured to confirm the role of *LaLBD37* in bulblet formation, including soluble sugar content, starch content, sucrose synthase activity, and endogenous hormone levels. The findings suggest that the *LaLBD37* gene plays a significant role in promoting the development of lily bulblets, offering insights for enhancing the reproductive capacity of Oriental hybrid lilies and exploring the molecular mechanisms involved in lily bulb regeneration.

## 1. Introduction

Oriental hybrid lilies are popular ornamental lilies, with rich colors, significant changes in flower patterns, long flowering periods, and diverse application methods [1]. However, in horticultural production, most Oriental lilies are propagated annually using new bulbs, resulting in high production costs and limiting their application in landscaping [2]. For instance, ‘Sorbonne’, a widely popular variety of Oriental lilies, is selected through hybrid breeding, necessitating the maintenance of many of its excellent traits through asexual reproduction. Traditional asexual reproduction methods such as bulb splitting and scale cutting have lower reproductive efficiency and longer cycles [3]. Therefore, cultivating new varieties of lilies with robust self-propagation ability is of great significance for expanding their application scope in gardens.

The regeneration of lily bulblets on scales essentially involves the direct regeneration of plant shoots. This process, known as organogenesis from scratch in plant regeneration, involves the emergence of adventitious roots and/or shoots from excised or injured plant organs [4]. Research has shown that external injury caused by peeling lily scales initially activates APETALA2/Ethylene Responsive Factor (AP2/ERF) transcription factors. This includes *LoERF109*, which is closely related to auxin metabolism pathways [5,6], and *LoERF115*, which confers cell division capability during organ regeneration [7]. After transcription, these transcription factors integrate various stress-related hormones throughout the developmental process, thereby activating cell division and leading to organ formation [8]. Additionally, several genes that play crucial roles in meristem initiation and are closely related to explant regeneration capability have been identified. These include ETHYLENE-RESPONSIVE TRANSCRIPTION FACTOR 2 (*LoESR2*), SHOOT MERISTEMLESS (*LoSTM*), WUSCHEL-RELATED HOMEOBOX 13-like (*LoWOX13*), CUP-SHAPED COTYLEDON3 (*LoCUC3*), HOMEOBOX PROTEIN KNOTTED-1-LIKE 6 (*LoKNAT6*), and HOMEOBOX GENE1 (*LoATH1*). These genes are significant for further targeted research on the robust organ regeneration capacity in lilies [9]. Additionally, in the formation of bulbils in lily stems, the genes Argonaute 1 (*LlAGO1)*, type-B RESPONSE REGULATORs (*LlRRs*), *LlWOX9/LlWOX11*, *LaKNOX1*, and *LaKNOX2* are considered to play important roles in promoting this process [10,11,12].

Carbohydrate metabolism is crucial for the growth and development of lily bulbs [13]. Starch, a polysaccharide, is a significant form of carbohydrate storage in lily bulbs, and the entire growth process of lilies is essentially a process of starch accumulation. Plant-soluble sugars, including sucrose, glucose, and fructose, are key components of sugar metabolism and transport. Soluble sugars serve as substrates for starch synthesis, with levels closely related to starch accumulation, and also directly affecting the nutritional status of the bulbs [14]. Sucrose, a disaccharide composed of one glucose molecule and one fructose molecule linked by a glycosidic bond, constitutes the majority of the total sugar content in lily bulbs, and changes in sucrose levels determine the variations in total sugar content. Many studies have investigated the starch or sucrose metabolism processes in lilies and other bulbous plants such as *Lycoris* [15,16]. These studies confirm that the metabolism of these two sugars has a decisive impact on the formation and enlargement of bulbs [17]. During the life cycle of lilies, assimilates produced through the photosynthesis of leaves undergo multiple transformations between “source” and “sink”, achieving morphological construction and nutrient storage [18]. In *Lycoris*, during the formation of new bulblets, the starch in the mother bulb is hydrolyzed by starch hydrolytic enzymes into soluble sugars, which provide nutrients for the growth of the new bulblets. Consequently, during the formation of new bulblets, the starch content in the mother bulb gradually decreases, while the starch content in the new bulblets gradually increases, along with an increase in soluble sugar content [19]. Studies have found that overexpressing the *LdSuSy* gene in the lily variety ‘White Heaven’ enhances sucrose synthase activity in the bulbs, thereby promoting carbohydrate accumulation and increasing starch deposition [20]. Therefore, the simultaneous increase in starch and soluble sugar content is regarded as a sign of the new bulblet expansion of lilies [20]. Meanwhile, the increase in sucrose synthase activity also contributes to an increase in starch and soluble sugar content.

Plant growth regulators play a crucial role in modulating plant growth and development, particularly in regulating bulb formation and development, which is a process driven by a complex hormone system [17]. Therefore, they are widely used in asexual reproduction of bulbous plants. In order to improve industrial efficiency, it is crucial to have a deep understanding of how plant hormones affect the development of lily bulbs. In this field, Auxin (IAA), Gibberellin (GA), and Abscisic acid (ABA) are considered important hormones that affect bulb development [21]. Auxin is the earliest discovered plant hormone, and indole acetic acid (IAA) is the main active component of natural plant auxin, which is a necessary plant hormone for cell division and enlargement [22]. Research has shown that the addition of Naphthyl acetic acid to the culture medium is beneficial for the formation of lily explant bulbs [23,24,25]. Gibberellin is a type of diterpenoid substance widely present in higher plants, which has the function of regulating plant growth and development [26]. The role of gibberellin in bulbs is currently controversial. In addition, gibberellin (GA) is thought to inhibit potato (*S. tuberosum*) tuber formation, at variance with its role in lilies [27]. However, there are also studies indicating that the synthesis and metabolism of starch are also influenced by endogenous gibberellins [28]. In the early stage of bulb enlargement, gibberellin helps to increase the activity of sucrose synthase, and the decrease in gibberellin content leads to a decrease in amylase activity, reducing starch consumption in the later stage of bulb enlargement. When using lily scales for cutting, gibberellin can reduce the decay rate of lily scales and increase the number of bulblets [21]. Abscisic acid is widely recognized as a promoting hormone in bulb formation. Some studies have shown that the inhibitory effect of GAs can be counteracted by ABA to promote bulb enlargement, while others have found that the antagonistic effect of ABA and GAs can regulate the accumulation of starch in *Gladiolus gandavensis*, thereby affecting its bulb development [29].

LATERAL ORGAN BOUNDARIES DOMAIN (LBD) proteins are plant-specific transcription factors with a conserved LOB domain, essential for regulating lateral organ development and metabolic processes, including anthocyanin and nitrogen metabolism [30]. The *LBD* gene family is divided into two categories: Class I and Class II. Previous studies have shown that Class I *LBDs* are mainly found at the edge of lateral organ primordia and play an important regulatory role in the growth of roots, leaves, flowers, and embryos. The Class II genes play an important role in regulating anthocyanin synthesis and nitrogen metabolism in plants [31,32,33,34,35,36,37,38]. In Arabidopsis, JAGGED LATERAL ORGANS (*AtJLO/LBD30*) regulates cell specification and organ patterning throughout plant development, and its dysfunction can lead to halted seedling development or altered organ initiation [39]. Two Arabidopsis LBD family members, AS1 ASYMMETRIC LEAVES 1 (AS1) and ASYMMETRIC LEAVES 2 (AS2), can inhibit KNOTTED1-LIKE HOMEOBOX (*KNOX*) expression by forming heterodimeric partners, thereby negatively regulating organ boundary establishment [40,41]. *AtASL4* (*AtLOB*) and *AtASL5* (*AtLBD12*) are involved in the development of meristem lateral organs [42,43], while *AtLBD29* (*AtASL16*), *AtLBD18* (*AtASL20*), and *AtLBD16* (*AtASL18*) are mainly expressed in lateral roots [44]. AtLBD18 and AtLBD33 form heterodimers, activating downstream gene *E2Fa* transcription, promoting the entry of root lateral meristem cells into the cell cycle, and affecting the development of *Arabidopsis* lateral roots [45,46]. In addition, a large number of studies on LBDs have been conducted in plants such as maize, rice, and wheat, all of which indicate that LBD is widely involved in regulating the development of lateral organs, thereby affecting crop yield [47,48,49,50]. Moreover, two Class I *LBD* genes, *LoLOB18* and *LlLBD18*, have been identified in lilies and are known to actively promote the development of bulbils and bulblets [51,52]. However, there is currently no research establishing a connection between Class II *LBD* genes and organ development in lilies. Based on early transcriptome data, the full-length *LaLBD37* gene was cloned. Its biochemical properties were assessed through subcellular localization and transcriptional activation assays. To further validate its function, an overexpression vector for *LaLBD37* was constructed and introduced into the Oriental lily ‘Sorbonne’. Positive plants were identified using GUS staining, PCR, and quantitative PCR to measure the relative expression levels of the target gene. Transgenic plants with higher expression levels were selected for phenotypic analysis. Finally, physiological changes in these plants, compared to the control group, were evaluated to investigate the function of the *LaLBD37* gene. This study provides both experimental and theoretical insights into the molecular mechanisms regulating lily bulblet regeneration.

## 2. Results

### 2.1. Full-Length Cloning and Sequence Analysis of LaLBD37

Based on the transcriptome annotation of the LaLBD37 (Isoform0013934) sequence, we cloned the full-length coding sequence of LaLBD37 (609 bp), which encodes 203 amino acids (Figure 1a). Amino acid sequence analysis indicated that LaLBD37 contains a LOB domain at the N-terminus (Figure 1b). Sequence alignment confirmed the conserved LOB domain at the N-terminus of LaLBD37 (Figure 1c). The phylogenetic tree of LBD37 from different species showed that LaLBD37 clusters with sequences from other monocots, with the highest similarity to the LBD37 amino acid sequence from Phoenix dactylifera (XP-008800648.1) (Appendix A).

### 2.2. Subcellular Localization and Transcriptional Activation of LaLBD37

To determine the subcellular localization of LaLBD37, the LaLBD37 cDNA was cloned into the pBI121-GFP vector and transformed into *Agrobacterium tumefaciens* strain GV3101, which was then used to infect *Nicotiana benthamiana* leaves. Following a one-day dark treatment, the plants were grown under long-day conditions for two days. Using the pBI121-GFP vector as a control, fluorescence microscopy was employed to observe the localization of the GFP-tagged LaLBD37. The nuclear localization of LaLBD37 was further confirmed by DAPI staining. In the control group, signals were detected in both the plasma membrane and the nucleus (Figure 2a). The recombinant plasmids of LaLBD37-A, LaLBD37-C, and LaLBD37-N, along with the pGBKT7 vector grew normally in SD/Leu/Trp-deficient medium after co-transformation with the p-GADT7 plasmid, indicating that the above plasmids had been successfully transferred into yeast cells (Figure 2b). However, only the yeast cells containing LaLBD37-A, LaLBD37-C, and pGBKT7-53 grew normally on SD/Leu/Trp/His/Ade-deficient medium. This result proves that the protein encoded by LaLBD37 has transcriptional activation activity and that the C-terminus contains the transcriptional activation domain. The pGBKT7 and pGADT7-T plasmids were co-transformed into yeast as a positive control, and the pGBKT7-Lamin and pGADT7-T plasmids were co-transformed into yeast as a negative control (Figure 2c). The above results show that LaLBD37 is a typical nuclear-localized transcription factor.

### 2.3. Acquisition of Resistant Plants

Using the prepared inoculum containing the pCAMBIA1300-*LaLBD37*-GUS plasmid, 300 sterile bulb scales of ‘Sorbonne’ were infected. Following pre-cultivation, co-cultivation, and selection with hygromycin (Figure 3a), a total of 45 scales were induced to bud (Figure 3b). After approximately 60 days of cultivation, the induced buds gradually developed into resistant plantlets (Figure 3c).

### 2.4. Identification of Genetically Modified Plants

#### 2.4.1. GUS Staining Identification

GUS staining of the leaves of resistant plants growing stably on hygromycin-containing medium showed that the leaves turned blue, indicating successful GUS gene expression during the transformation of ‘Sorbonne’ adventitious buds (Figure 4a). While this preliminary result suggests successful plasmid transfer, further validation is needed to confirm stable expression.

#### 2.4.2. DNA Level Identification

PCR validation was performed on the resistant plants that exhibited blue staining in the previous GUS activity test. The results indicated that the transgenic resistant plants of *LaLBD37* produced specific fragments of approximately 600 bp, which were identical to those of the vector plasmid. Consequently, L1–L8 were identified as transgenic plants, but the intensity of the fragments obtained from different lines varied. It is speculated that this difference in expression level was caused by different copy numbers of genes in different transgenic offspring (Figure 4b), indicating that the target gene has been successfully transferred into the genome of ‘Sorbonne’.

#### 2.4.3. RNA Level Identification

Leaves from the transgenic plants identified in the previous step were harvested, and total RNA was extracted for real-time quantitative PCR analysis. The expression levels were compared to those of the control plants (ck) transformed with the empty vector (pCAMBIA1300-GUS vector). In the *LaLBD37* transgenic lines, the expression levels of the transgenic plants were significantly higher than those of the control. Specifically, the expression levels in plants L1, L2, and L4 were 16.95, 15.60, and 11.54 times higher than the control, respectively (Figure 4c). The real-time quantitative PCR results further confirmed that the previously screened and validated transgenic plants were positive. For subsequent experiments, the three lines with the highest relative expression levels, L1, L2, and L4, were selected from the *LaLBD37* overexpression plants.

### 2.5. Morphological Changes in Transgenic Lilies

Three transgenic plants of *LaLBD37* with the highest expression levels were selected for morphological comparison with the control plants under the same culture conditions. The results showed that all the transgenic ‘Sorbonne’ lily plants had an increase in the number of scales and more plump scales (Figure 5a). From a data perspective, in the transgenic plants of *LaLBD37*, the number of scales significantly increased by 55.37% compared to control plants (ck) (Figure 5b). These preliminary findings suggest that the *LaLBD37* gene has a promoting effect on lily bulb development.

### 2.6. Changes in Soluble Sugar, Starch Content, and Sucrose Synthase Activity in Transgenic Lines

Lily bulbs store soluble sugars, impacting bulblet formation, development, and overall plant growth. This study analyzed the soluble sugar content in transgenic and control plants. Results revealed a significant increase in soluble sugar content in the bulblets of *LaLBD37* transgenic plants. Specifically, the overexpressed plants exhibited a notable rise in soluble sugars, with content increasing by 90.71%, 64.51%, and 45.99% in the L1, L2, and L4 lines, respectively (Figure 6a). Additionally, starch content was also measured in transgenic and control plants, showing a significant increase in *LaLBD37* transgenic plants. The starch content displayed an upward trend in the transgenic plants, with the L1, L2, and L4 lines exhibiting increases of 80.08%, 78.08%, and 58.76% in starch content, respectively (Figure 6b). Furthermore, the activity of sucrose synthase (decomposition direction) was measured in the same period scales. In the *LaLBD37* transgenic lines, there was a significant increase in the activity of sucrose synthase in the bulbs. The activity in the L1, L2, and L4 lines increased by 93.41%, 48.27%, and 30.21%, respectively (Figure 6c), aligning with the trend observed in soluble sugar content.

### 2.7. Determination of Endogenous Hormone Content in Transgenic Lines

To determine the endogenous hormone levels in various transgenic lines, we selected three transgenic lines and one control line at the same developmental stage for experimentation. The content of IAA in the bulblets of the *LaLBD37* overexpression lines was significantly higher than that in the empty vector control line. Specifically, the IAA levels in the L1, L2, and L4 lines were 2.40, 1.88, and 2.97 times that of the control (ck) line, respectively (Figure 7a). When measuring the endogenous GA3 levels in the bulbs at the same stage, the results indicated that there was no unidirectional linear relationship between the *LaLBD37* overexpression lines and the control line. The GA3 levels in L1 and L4 were markedly higher than those in the control, whereas the GA3 level in L2 was significantly lower. The GA3 content in the bulbs of the L1, L2, and L4 lines was 1.35, 0.60, and 1.34 times that of the control, respectively (Figure 7b). Moreover, during the same stage, the ABA content in the bulbs of the *LaLBD37* overexpression lines was significantly elevated compared to the empty vector control line. Specifically, the ABA levels in L1, L2, and L4 were 4.35, 6.23, and 8.19 times that of the control, respectively (Figure 7c).

## 3. Discussion

LBD proteins constitute a plant-specific family of transcription factors characterized by a highly conserved LOB domain, which plays crucial roles in the regulation of lateral organ development and metabolic processes [30]. The LOB domain comprises specific sequences, including a CX2CX6CX3C zinc finger-like structure, a GAS region, and an LX6LX6LX6LX6 leucine zipper motif [53]. Through previous transcriptomic analysis, we identified a member of the LBD family named *LaLBD37*. Analysis using the SMART conserved domain database revealed that the protein sequence encoded by *LaLBD37* contains a conserved LOB domain at its N-terminus. Therefore, we believe that LaLBD37 belongs to the LBD transcription factor family. Transcription factors typically function within the nucleus to regulate gene expression. Lee et al. [54] provided robust evidence supporting the notion that the LBD family comprises transcription factors by demonstrating that proteins encoded by LOB are localized in the nucleus. Li et al. [48] used the D53-m Cherry nuclear localization marker to transiently transform rice protoplasts with OsLBD37 and OsLBD38 linked to the pA7-GFP vector, revealing that both genes are localized in the nucleus. Similarly, in *Brassica*, the protein encoded by BcLBD37 is also nuclear [55]. However, not all LBD family members are exclusively nuclear. For instance, Arabidopsis AtLBD10 is found in both the nucleus and cytoplasm, AtLBD22 is cytoplasmic, and AtLBD27 and AtLBD36 are primarily cytoplasmic with weak nuclear signals [33,56]. In lilies, LlLBD18 is present in both the nucleus and cytoplasm [52]. Our experiments show that LaLBD37 is exclusively expressed in the nucleus of *Nicotiana benthamiana* leaves, aligning with its role as a transcription factor and suggesting it likely regulates gene expression by interacting with nuclear target elements. In rice, OsLBD37 and OsLBD38 exhibit transcriptional activation activity [48]. The protein encoded by the lily *LaLBD37* gene also demonstrates transcriptional activation in a yeast system, with its C-terminal region being crucial for this activity. These findings collectively indicate that LaLBD37 is a typical nuclear-localized transcription factor involved in plant growth and development, with its C-terminal domain providing transcriptional activation or repression functions to target genes [34]. 

Lily bulblet occurs as a result of the activation of adventitious meristematic tissue at the base of the scales [57]. Bell et al. found that aberrant expression of the *AtLOB* gene activated the *BAS1* gene encoding the oleuropein lactone (BR) inhibitory enzyme, which reduced the plant response to BR, and the LOB and BR signals formed a feedback loop to regulate the accumulation of BR in the organ boundary and ultimately inhibited the growth of organ boundary cells [32]. Overexpression of the rice *OsAS2* (*LBD6*) gene, which is expressed in the apical meristem, leaf primordia, and young leaves, inhibits stem differentiation, promotes cell division, and delays healing tissue cell differentiation in rice [58]. In *L. lancifolium* and Oriental Lily ‘Siberia’, the *LBD16* gene has been shown to promote bulbil formation in different varieties by gene silencing techniques [51,52]. Meng et al. found that class II LBD family members also affect the construction of plant growth axis, and overexpression of *AtASL38/AtLBD41* in *Celosia cristata* caused the dorsal-ventral torsion of leaf blades, which resulted in the formation of extremely curled leaf blades [59]. In Arabidopsis, aberrant expression of the LBD family members *AtASL1/AtLBD36* causes errors in the construction of the acropetal axis. Given that *LaLBD37* is a homolog of Arabidopsis type II *LBD* genes, we hypothesized that a similar effect might occur on lateral organs, i.e., LaLBD37 could increase the activity of indeterminate meristematic tissues, thereby accelerating their differentiation and leading to the formation of a greater number of bulblets. Therefore, we hypothesized that *LaLBD37* is involved in regulating the development of lily bulb scales, potentially enhancing bulblet reproductive capacity.

The bulb of a lily is a specialized organ formed by the enlargement of underground stems and axillary buds, serving both reproductive and nutritional functions [2]. Its formation and enlargement are regulated by various factors and are a very complex physiological process. Carbohydrates play a crucial role in the growth, development, and physiological activities of bulbs. The storage of nutrients in bulbs mainly relies on starch. When the bulblets regenerate and expand in the scales, starch will be decomposed, leading to a decrease in the starch content of the maternal scales. Through transcriptome analysis of different developmental stages of bulbs, it was found that enzymes involved in starch synthesis, such as soluble starch synthase (SSS), ADPG pyrophosphorylase (AGPase), and granular-bound starch synthase (GBSS), have higher gene activity in bulblets, while they show a downward trend in maternal scales [13,60]. In bulblets, starch in the maternal scale is hydrolyzed into soluble sugars by starch hydrolases, providing nutrients for the growth of new bulbs. Therefore, during the formation of new bulbs, the starch content in the maternal scale gradually decreases, while the starch content in bulblets gradually increases, and the soluble sugar content also gradually increases. Just like in the study of tulips, the simultaneous increase of soluble sugar content and amylase activity in scales marks the initiation of flower bud differentiation [61]. In this experiment, the soluble sugar and starch content in the scales of transgenic plants were significantly higher than those in the control group. At the same time, genetically modified lily scales exhibited an increase in number and enlargement in size. This variation parallels the changes in a series of physiological indicators related to the transformation of sink–source relationship during the development process of lily bulbs. The generation and decomposition processes of starch are closely related to the metabolism of sucrose, which directly affect the bulbs and overall growth of lilies. 

Hormones play crucial regulatory roles in the growth and development of plants, and their effects are not solely attributable to individual hormones but rather to the coordinated interactions among endogenous hormones [62]. Research has shown that auxin alone or in conjunction with other hormones plays a significant role in regulating plant responses to environmental stimuli [63]. In *Lilium lancifolium*, an increase in endogenous IAA levels has been observed during bulbil formation [64], a finding also demonstrated in *Lycoris radiata* [65]. In our study, the endogenous IAA content of overexpressed *LaLBD37* plants was significantly higher than in the controls. We hypothesized that within a certain range, high expression of IAA content could promote bulbogenesis. Gibberellin can stimulate seed germination, stem elongation, leaf unfolding, and the development process of flowers. Meanwhile, gibberellin also helps to increase plant height, internode spacing, and leaf fresh weight, thereby increasing the overall biomass of plants [66]. Recent studies have demonstrated an increase in endogenous gibberellin content during the formation of bulblets from mother scales [67]. Conversely, in lilies, additional research has shown that applying 6-BA externally promotes bulb formation, yet all four endogenous gibberellin forms (GA1, GA3, GA4, and GA7) exhibit varying degrees of decline during bulb formation [12]. In this experiment, the detection of GA3 content showed that the L1 and L4 contents were higher than the control group, while the GA3 content of L2 was lower than the control group. The effect of GA3 on the occurrence of lily bulbs is not entirely clear and does not follow a linear relationship. The occurrence of such a situation is not only consistent with previous research indicating the controversial role of gibberellins on bulbs, but also possibly due to the inaccurate determination of gibberellins during tissue culture seedlings. Therefore, further detailed research can investigate how gibberellins regulate bulblet formation following seedling refinement. In lilies, abscisic acid (ABA) is well-recognized for its role in promoting bulb formation [68]. It was found that a certain concentration of ABA could stimulate the formation and enlargement of lily bulbs [69]. Similarly, during bulb expansion in *Acorus calamus*, ABA content increased dramatically, accompanied by an increase in starch content, which in turn favored the expansion of storage organs [29]. In tuberous plants, such as potato and taro, a rise in ABA content also accompanies tuber formation [70,71]. In this experiment, by detecting the ABA content in the scales of pCAMBIA1300-*LaLBD37*-GUS and pCAMBIA1300-GUS transgenic lily plants at the same period, the results showed that the ABA content of the lilies transfected with the *LaLBD37* gene was significantly higher than that of the control group, which is consistent with the results of previous studies.

## 4. Materials and Methods

### 4.1. Plant Materials

Using the ‘Aladdin’ leaves of lilies grown in nurseries as raw materials, RNA was extracted and gene cloning of *LaLBD37* was performed. Aseptic seedlings were obtained through the adventitious bud regeneration pathway using the oriental lily ‘Sorbonne’ bulbs as explants, and their plump and healthy small scales were taken as genetic transformation receptor materials.

### 4.2. Gene Cloning and Analysis of LaLBD37

Based on transcriptome sequence information, full-length amplification primers for the lily *LaLBD37* gene were designed using the SnapGene software package https://www.snapgene.com accessed on 15 April 2024. Using lily leaves as materials, total RNA was extracted using the polysaccharide polyphenol RNA rapid extraction kit (Aidlab, Beijing, China), and reverse transcribed into cDNA using the Trans Script One Step g DNA Removal and cDNA Synthesis Super Mix kit (Aidlab, Beijing, China). PCR amplification was performed using the ‘Aladdin’ cDNA of lilies as a template. Conserved protein structural domains were analyzed using SMART (http://smart.embl.de/ (accessed on 15 April 2024)). Multiple sequence comparisons were analyzed using the DNAMAN package v10. Phylogenetic analyses were performed using MEGA6 (http://mega6.software.informer.com/ (accessed on 2 November 2023)).

### 4.3. Subcellular Localization of LaLBD37

The *LaLBD37* full-length cDNA, controlled by the 35S cauliflower mosaic virus promoter, was inserted into the pBI121-GFP vector utilizing the ClonExpress^®^ II One Step connecting reagent kit (Vazyme, Nanjing, China). The primer pair sequences for amplification can be found in Appendix A. Subsequently, the resultant plasmid was introduced into Agrobacterium tumefaciens strain GV3101. Agrobacterium cells were harvested and suspended in an infiltration buffer containing 10 mmol/L MgCl2, 10 mmol/L MES, and 150 μmol/L As, with an optical density (OD 600) of 1.0, then infiltrated into *Nicotiana benthamiana* leaves. Approximately three days post-infiltration, the leaves were harvested and stained with DAPI (Coolaber, Beijing, China). Imaging was performed using a Leica TCS SP8 confocal scanning microscope, with GFP and DAPI fluorescence observed under 488 nm and 340 nm excitations, respectively.

### 4.4. Detection of Transcriptional Autoactivation Activity in LaLBD37

The yeast GAL4 two-hybrid system was used to detect transcriptional activation. The full-length cDNA sequence of LaLBD37 and the truncated sequences of LaLBD37 (LaLBD37-N and LaLBD37) were fused to the coding sequence of the GAL4 binding domain and transformed into yeast strain AH109. pGBKT7 plasmid and pGADT7-T plasmid were co-transformed into yeast as the positive control, and pGBKT7-Lamin plasmid and pGADT7-T plasmid were co-transformed into yeast as the negative control. The pGBKT7 plasmid and pGADT7-T plasmid were used as a positive control, and pGBKT7-Lamin plasmid and pGADT7-T plasmid were used as a negative control. The primers are listed in Appendix A.

### 4.5. Agrobacterium Mediated Transformation of Small Scales from Sterile Seedlings of ‘Sorbonne’

*LaLBD37* full-length cDNA was constructed into the pCAMBIA1300-GUS vector. The constructed pCAMBIA1300-*LBD37*-GUS and pCAMBIA1300-GUS plasmids were separately transformed into small scales by Agrobacterium tumefaciens GV3101-mediated transformation.

Healthy and plump small scales were peeled off from sterile seedlings of ‘Sorbonne’ and placed in a scale induction medium (MS + 0.5 mg/L 6-BA + 1 mg/L NAA + 30.0 g/L sucrose + 7.0 g/L agar). This was pre-cultured for 3 days under dark conditions at 25 ± 2 °C. The small scales of sterile seedlings of ‘Sorbonne’ were infected with a prepared infection solution containing pCAMBIA1300-*LaLBD37*-GUS plasmid, with an OD600 value of 0.7 and an infection time of 15 min. After infection, to co-culture was transferred and dark cultured for 3 days. The co-culture medium for small scales of sterile seedlings in ‘Sorbonne’ was: MS + 0.5 mg/L 6-BA + 1 mg/L NAA + 100 μ Mol/L AS + 30.0 g/L sucrose + 7.0 g/L agar. The co-cultured receptor materials were then transferred into the screening medium. The screening medium for sterile seedling scales of ‘Sorbonne’ was MS + 0.5 mg/L 6-BA + 1 mg/L NAA + 30.0 g/L sucrose + 7.0 g/L agar + 50 mg/L Hg + 300 mg/L Cef. Sterile seedling scales were cultured under light for 8 h/16 h and screened for 60 days at 25 ± 2 °C. Finally, the obtained resistant adventitious buds were inoculated into MS medium and cultured for about 2 months until the grown resistant seedlings were obtained. These seedlings will be used for later transgenic plant testing.

### 4.6. Identification of Genetically Modified Plants

#### 4.6.1. Detection of GUS Activity in Transformed Plants

The leaves of the obtained resistant seedlings were cut into small pieces and placed into a 10 mL centrifuge tube, to which a sufficient amount of pre-prepared GUS stain was added to ensure that the sample was completely covered by the stain. The centrifuge tubes containing the samples and the staining solution were then placed in a vacuum apparatus at a pressure of 0.1 MPa for 30 min. The tubes were then left to stain for 24 h at a constant temperature of 37 °C. The samples were removed from the tubes and stained with a 70% solution. Finally, the samples were removed and destained 2–3 times with 70% ethanol, and then the color change of the leaves was observed under a light microscope.

#### 4.6.2. PCR Detection of Transformed Plants

The ‘Sorbonne’ lily leaves used for the assay were cut, rapidly cooled, and ground in liquid nitrogen. DNA was then extracted using the CTAB Plant Genome DNA Rapid Extraction Kit (Aidlab, Beijing, China). Following DNA extraction, the DNA concentration was measured, and the target genes were amplified by PCR. The gene cloning primers *LaLBD37*-F and *LaLBD37*-R from Appendix A were used as PCR validation primers to detect pCAMBIA1300-*LaLBD37*-GUS resistant seedlings. Templates included untransformed ‘Sorbonne’ lily leaves, leaves of resistant seedlings transformed with pCAMBIA1300-GUS, water, and DNA from plasmids pCAMBIA1300-GUS and pCAMBIA1300-*LaLBD37*-GUS resistant seedling leaves. PCR products were analyzed by agarose gel electrophoresis to confirm the presence of transgenic plants.

#### 4.6.3. Real-Time Fluorescence Quantitative Detection of Transformed Plants

Total RNA was extracted from the scales of pCAMBIA1300-*LaLBD37*-GUS plants and pCAMBIA1300-GUS plants at the same developmental stage and reverse transcribed into cDNA. Fluorescent quantitative primers were designed using Primer Premier 5.0 software, with *LiTIP41* as an internal reference gene. The primer sequences are shown in Appendix A. Reactions were performed on a CFX96 Real-Time System instrument (Bio-Rad, Hercules, CA, USA) according to the instructions provided with the Taq Pro Universal SYBR qPCR Master Mix (Vazyme, Nanjing, China). The amplification procedure was as follows: pre-denaturation at 95 °C for 2 min, denaturation at 95 °C for 10 s, and annealing at 60 °C for 1 min, for a total of 45 cycles. Data were analyzed using the 2^−ΔΔCt^ method. The relative expression levels of the target genes in the scales of the overexpression lines were measured by real-time quantitative PCR, using pCAMBIA1300-GUS resistant plants as the control. All RT–qPCR assays were performed with three biological replicates.

### 4.7. Phenotypic Observation of Transgenic ‘Sorbonne’ Lines

The screened transgenic pCAMBIA1300-*LaLBD37*-GUS lily seedlings and transgenic pCAMBIA1300-GUS lily seedlings were transferred to MS medium and cultured for two months, with the medium changed every 20 days. Morphological differences between the two groups, particularly in the number and size of scales, were observed.

### 4.8. Metabolite Analysis

Bulblets from pCAMBIA1300-*LaLBD37*-GUS and pCAMBIA1300-GUS transgenic lily plants were collected simultaneously and stored at −80 °C. Changes in soluble sugar content and starch content in the bulblets were measured using the Plant Soluble Sugar Content Assay Kit and the Plant Starch Content Assay Kit, respectively. Subsequently, sucrose synthase activity in the bulblets of both transgenic plant groups was measured using the Sucrose Synthase Activity Assay Kit (Keming, Suzhou, China). Each sample was tested in triplicate.

### 4.9. Determination of Endogenous Hormone Content in Transgenic Lines

Bulblets at the same developmental stage from pCAMBIA1300-*LaLBD37*-GUS and pCAMBIA1300-GUS transgenic lily plants were selected and ground in liquid nitrogen. A precise amount of 0.2 g of the ground powder was placed into a 2 mL centrifuge tube. The levels of three endogenous hormones—indole-3-acetic acid (IAA), gibberellic acid (GA3), and abscisic acid (ABA)—were measured using specific enzyme-linked immunosorbent assays (ELISAs). Each measurement was repeated three times for accuracy.

### 4.10. Data Analysis

The measured physiological indicators and endogenous hormone data were organized and classified using Microsoft Excel 2013. Then, one-way ANOVA was performed on the organized data in SPSS 26.0 (SPSS, Chicago, IL, USA), and graphics were drawn using Prism 9.0 and Photoshop 2024.

## 5. Conclusions

In this study, the gene *LaLBD37*, associated with bulblet development, was cloned from the ‘Aladdin’ LA lily using bioinformatics analysis. It was confirmed to be a transcriptional activator localized in the nucleus and classified as a member of the *LBD* class II family. Subsequently, early-cultured ‘Sorbonne’ sterile bulbs were used as explants for genetic transformation. After obtaining resistant plants, three transgenic lines were identified through GUS staining, DNA, and RNA analyses. Phenotypic observations indicated that these three lines exhibited significantly higher numbers of scales and bulb volumes compared to the control. Additionally, the soluble sugar content, starch content, and sucrose synthase activity were elevated in the transgenic lines. The levels of endogenous hormones (IAA, ABA) were also significantly higher. These results suggest that *LaLBD37* may enhance organ primordia initiation at the base of the bulb and regulate bulb development by interacting with carbohydrate and hormone metabolic pathways. We anticipate that our research will enhance lily bulblet regeneration efficiency, enabling the production of more and larger bulblets within the same timeframe. This improvement is expected to facilitate large-scale production and commercialization, reduce costs, and meet market demands, ultimately promoting the widespread cultivation and application of lilies.

## Figures and Tables

**Figure 1 ijms-25-09456-f001:**
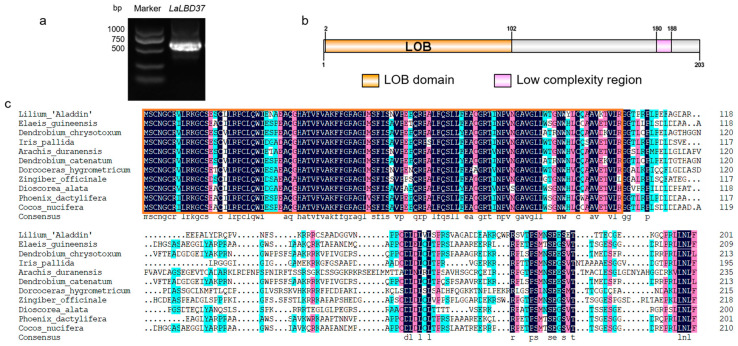
Full-length cloning and sequence alignment of LaLBD37. (**a**) Full-length cloning and domain prediction of LaLBD37; (**b**) domain prediction of LaLBD37; (**c**) Multiple sequence alignments of the amino acids of LaLBD37.

**Figure 2 ijms-25-09456-f002:**
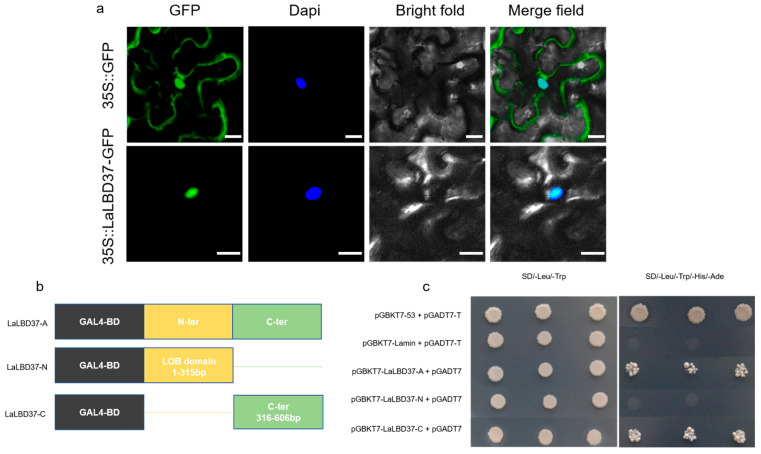
Analysis of subcellular localization and transcriptional activation activity of LaLBD37. (**a**) Nuclear localization of LaLBD37; (**b**) The division of N- and C-terminal regions of LaLBD37; (**c**) Transcriptional activation assay. Scale bar, 20 µm.

**Figure 3 ijms-25-09456-f003:**
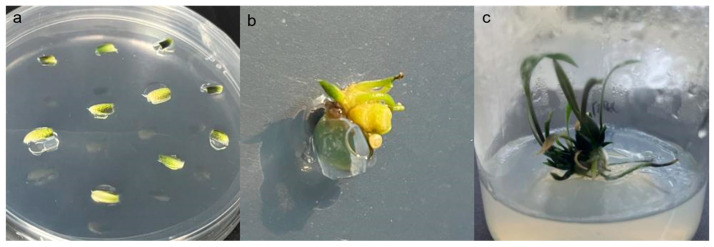
Growth process of transgenic plants. (**a**) Screening culture process; (**b**) Germination process; (**c**) Growth process.

**Figure 4 ijms-25-09456-f004:**
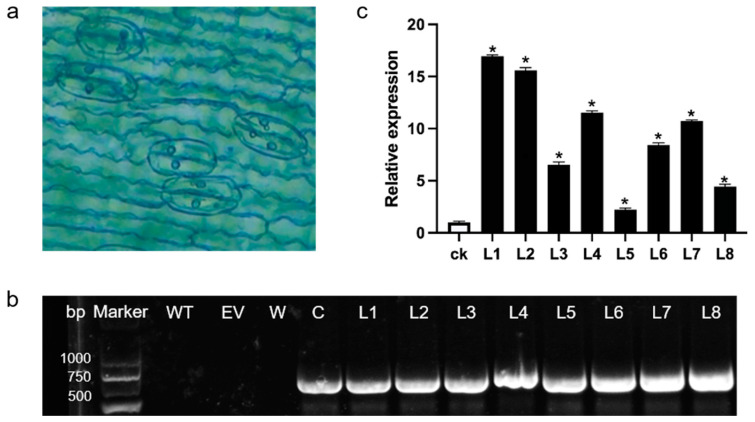
Validation of transgenic ‘Sorbonne’ seedlings. (**a**) GUS detection in ‘Sorbonne’ transgenic *LaLBD37* resistant plants; (**b**) PCR detection of ‘Sorbonne’ resistant plants. WT: Untransformed ‘Sorbonne’ plants; EV: Empty vector-transformed ‘Sorbonne’ plants; W: Water; C: Carrier vector; (**c**) Quantitative analysis of *LaLBD37* in ‘Sorbonne’ positive plants; each group of data was the mean of three repetitions; ck: empty vector-transformed ‘Sorbonne’ line; L1–L8: *LaLBD37*-transformed ‘Sorbonne’ lines; * represents significant difference (*p* < 0.05).

**Figure 5 ijms-25-09456-f005:**
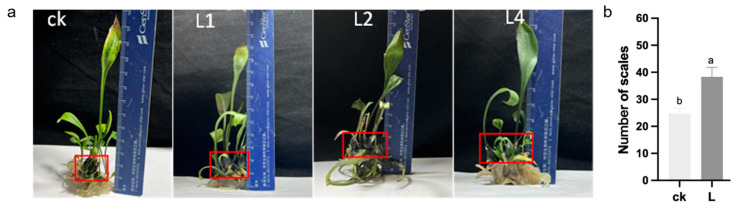
Phenotypic observation and scale count of transgenic ‘Sorbonne’. (**a**) Phenotypic observation of transformed empty vector ‘Sorbonne’ and transgenic ‘Sorbonne’. ck: empty vector-transformed ‘Sorbonne’; L1, L2, and L4: lines with high expression of *LaLBD37* in ‘Sorbonne’; (**b**) Number of scales for the transforming empty vector ‘Sorbonne’ and transgenic ‘Sorbonne’. ck: empty vector-transformed ‘Sorbonne’ line; L: *LaLBD37*-transformed ‘Sorbonne’ lines; Different letters represent significant differences among the three groups.

**Figure 6 ijms-25-09456-f006:**
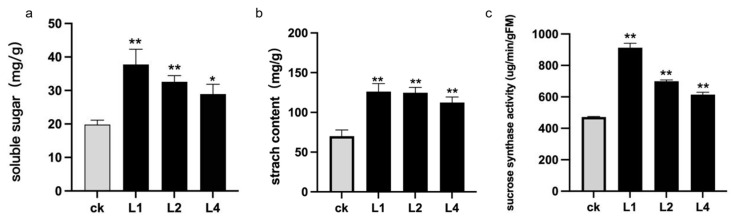
Phenotypic observation and scale count of transgenic ‘Sorbonne’. (**a**) The change of soluble sugar content in *LaLBD37* transgenic lines; (**b**) The change of starch content in *LaLBD37* transgenic lines; (**c**) The change of sucrose synthase activity in *LaLBD37* transgenic lines; each group was the average of three repetitions; ck: empty vector-transformed ‘Sorbonne’; L1, L2, and L4: lines with high expression of *LaLBD37* in ‘Sorbonne’; * representing significant difference (*p* < 0.05); ** represents significant difference (*p* < 0.01).

**Figure 7 ijms-25-09456-f007:**
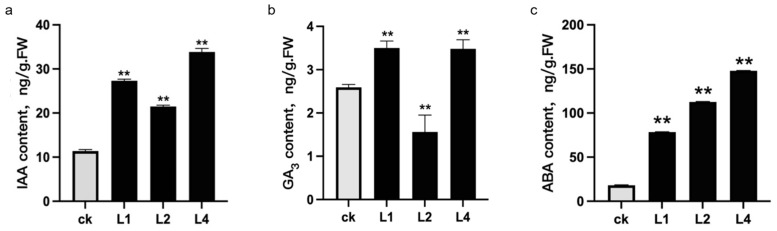
Determination of endogenous hormone content in *LaLBD37* overexpression lines at the same time. (**a**) The IAA content in *LaLBD37* transgenic lines; (**b**) The GA3 content in *LaLBD37* transgenic lines; (**c**) The ABA content in *LaLBD37* transgenic lines; each group was the average of three repetitions; ck: empty vector-transformed ‘Sorbonne’; L1, L2, and L4: lines with high expression of *LaLBD37* in ‘Sorbonne’; ** represents significant difference (*p* < 0.01).

## Data Availability

Data are contained within the article and Appendix A.

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
