# Peer review of "Functional Study on the Key Gene LaLBD37 Related to the Lily Bulblets Formation"

_ijms, 2024, doi:10.3390/ijms25179456_

Round 1

Reviewer 1 Report

Comments and Suggestions for Authors

The work regards to the higher bulb production, I think. However, the genes were studied and, I think, the really aim of this research should be highlighted including the application possibility.

The work should be read carefully because some grammatically, spaces and font mistakes are present, like in line 35, 337, 344 etc.. And a few somewhat unclear statements, like in abstract, and in lines 41-47.

The work should have hypotheses and conduct to their explanation, support or rebut. I have a problem with comparing the abstract / introduction and conclusions…

Lines 119-140: this part is to long and unclear. Please sort out and explain. It is possible that division is a good solution.

Line 13: natural environment? No. The optimal conditions in cultivation, I think.

Line 35: variety?

Line 61: 'White heaven' lilies?

Line 352: ?

Line 355: italics: be carefully, the latin names should be written in italics in all the text.

Line 373, 415: ?

Line 499-500: ? Rethink it and form in another way, please.

Line 509-516: ?

Comments on the Quality of English Language

The language seems to be a bit careless.

Author Response

Please see the attachment:

Response letter

Journal: IJMS (ISSN 1422-0067)

Manuscript: ijms-3155983

Title: Functional study on the key gene LaLBD37 related to the Lily bulblets formation

Dear reviewer:

Thank you for providing us with an opportunity to improve the quality of the submitted manuscript (ijms-3155983). We greatly appreciate the constructive and insightful comments of the reviewers. In this revision, we have responded to all of these comments and have made extensive touches to the English language to ensure a smooth read. We hope that the revised manuscript will meet the publication standards of your journal. The following is our point-by-point response to your comments. In the revised manuscript, all changes are highlighted in red.

Comment 1: The work regards to the higher bulb production, I think. However, the genes were studied and, I think, the really aim of this research should be highlighted including the application possibility.

ResponseThank you very much for your suggestion. In the manuscript we add the application possibilities of the study in the conclusion section. The current lily transgenic system is not mature enough, and being able to improve it by obtaining genetically stable transgenic LaLBD37 lily seedlings is something that could contribute to the perfection of the lily transgenic system. Phenotypic statistical observation found that compared with the control, the transgenic strain of the number of scales and bulb volume increased, so we believe that the LaLBD37 gene is functional, which can promote the development of organ primordia and bulb formation, but the specific molecular mechanisms need to continue to be further investigated. Thus, in the future, in terms of lily bulb propagation, if it can be put into the industry through tissue culture, it will improve the efficiency of lily propagation, which is the possibility of application and a good vision.

Comment 2: The work should be read carefully because some grammatically, spaces and font mistakes are present, like in line 35, 337, 344 etc. and a few somewhat unclear statements, like in abstract, and in lines 41-47.

ResponseThanks for your suggestion. We have corrected all grammatical, space and font errors in the manuscript and have cut lines 41-47 from the original. Better writing is also shown in the manuscript (page 1, line 37-40).

Comment 3: The work should have hypotheses and conduct to their explanation, support or rebut. I have a problem with comparing the abstract / introduction and conclusions

ResponseThank you for your comments. We have made assumptions and answered them in the introduction and discussion sections of the manuscript. Because of the major changes made to the manuscript, a detailed enumeration has not been made, but rather most of the content of the individual chapters has been revised.

Comment 4: Lines 119-140: this part is too long and unclear. Please sort out and explain. It is possible that division is a good solution.

ResponseThank you very much for your comment. In the last paragraph of the Introduction, we have summarised and distilled the main experimental content of the whole manuscript and briefly stated the value for lily breeding to make it more concise (page 3, line 143-152).

Comment 5: Line 13: natural environment? No. The optimal conditions in cultivation, I think.

ResponseThanks for your suggestion. We strongly agree with your comments and have made changes in the manuscript (page 1, line 13).

Comment 6: Line 35, line 415: variety?

ResponseWe are very sorry that the language caused you distress, we have rewritten the sentence and explained to you that ’Sorbonne’ is a variety of Oriental Lily.

Comment 7: Line 61: 'White heaven' lilies?

ResponseThanks for your question. ’White heaven’ is a variety of Lilium longiflorum.

Comment 8: Line 352: ?

ResponseThank you very much for your meticulous reading of the manuscript. We found an error in this sentence and corrected it (page 10, line 412-414).

Comment 9: Line 355: italics: be carefully, the latin names should be written in italics in all the text.

ResponseThank you very much for your advice. We have written gene and species names mentioned throughout the manuscript in italics, while proteins are written in block letters. Please forgive me if I don't list them all for you because the changes to this manuscript are huge.

Comment 10: Line 373: ?

ResponseThank you very much for your question, we have corrected it in the manuscript (page 11, line 434).

Comment 11: Line 499-500: ? Rethink it and form in another way, please. Line 509-516: ?

ResponseThank you very much for your comments, we have rewritten the conclusion of this article, in addition to summarising all the experimental results presented in the manuscript, we also focus on describing the importance of a stable lily genetic transformation system for lily breeding for the lily industry and the new findings of the LaLBD37 transcription factor in the proliferation of lily bulblets (page 13, line 545-562).

Best regards

Sincerely yours

Yingmin Lyu

Reviewer 2 Report

Comments and Suggestions for Authors

Understanding the genetic basis of lily bulb formation will increase our knowledge in this area. This is a very timely topic that can be applied to other bulb crops. The manuscript also deals with the issue of genetically modified organisms. Unfortunately, there is mention of abnormality formation in the text within the results, but nowhere is it explained why this occurs. The abstract is adequate. Some passages in the introduction are quite familiar. This is mainly the section on phytohormones, but on the other hand this section should be expanded with regard to tuber formation. On line 52 it is stated that starch and sucrose are sugars. In my opinion, a distinction should be made between simple sugars and polysaccharides. Again, there is no passage dealing with soluble sugars. The methodology is adequate. The results are well described except for the chapter on phytohormones. Values are given in %, but these are not shown in the graphs. Given the references to the graphs, the description should be uniform. For graphs 6 and 7, the axis descriptions are missing. The discussion is descriptive in places. The citation of individual sources needs to be double-checked. It is inconsistent. 

Author Response

Please see the attachment:

Response letter

Journal: IJMS (ISSN 1422-0067)

Manuscript: ijms-3155983

Title: Functional study on the key gene LaLBD37 related to the Lily bulblets formation

Dear reviewer:

Thank you for providing us with an opportunity to improve the quality of the submitted manuscript (ijms-3155983). We greatly appreciate the constructive and insightful comments of the reviewers. In this revision, we have responded to all of these comments and have made extensive touches to the English language to ensure a smooth read. We hope that the revised manuscript will meet the publication standards of your journal. The following is our point-by-point response to your comments. In the revised manuscript, all changes are highlighted in red.

Comment 1: Unfortunately, there is mention of abnormality formation in the text within the results, but nowhere is it explained why this occurs.

Response: Thanks to your comments. We explored the possibility of increased bulblet size and number of scales in the discussion section of the manuscript (page 9, line 360-363).

Comment 2: Some passages in the introduction are quite familiar. This is mainly the section on phytohormones, but on the other hand this section should be expanded with regard to tuber formation.

ResponseThanks for your advice. We have added the factors of bulblet formation and the mode of action of some of the genes that play a regulatory role in the Introduction section, as well as revising the section on phytohormones (page 2, line 47-65).

Comment 3: On line 52 it is stated that starch and sucrose are sugars. In my opinion, a distinction should be made between simple sugars and polysaccharides. Again, there is no passage dealing with soluble sugars.

ResponseThanks for your question. We have made the nature of sucrose and starch more explicit in the manuscript and have overhauled the carbohydrate metabolism paragraph in the Introduction to add a section on soluble sugars (page 2, line 66-90).

Comment 4: The results are well described except for the chapter on phytohormones. Values are given in %, but these are not shown in the graphs. Given the references to the graphs, the description should be uniform. For graphs 6 and 7, the axis descriptions are missing.

ResponseThank you very much for your valuable input. We have re-corrected the text in the plant hormones section and optimised the language. Regarding your comment about displaying specific values, we did not add them to the graphs to maintain uniformity, as they are not shown in the other graphs. And the descriptions are in multiples that more visually show the differences. In addition, we have added the coordinate annotations for Figures 6 and 7.

Comment 5: The discussion is descriptive in places.

ResponseThanks for your question. We have greatly adapted the discussion section to explore the possible function of the LaLBD37 gene in lily bulblet development by combining the results of the present experiment with other previous studies (page 9, line318).

Comment 6: The citation of individual sources needs to be double-checked. It is inconsistent. 

ResponseThank you for your comments. We have made a complete and detailed check and revision of the citation in the manuscript to ensure that there are no more errors that cannot be corresponded to.

Best regards

Sincerely yours

Yingmin Lyu

Round 2

Reviewer 1 Report

Comments and Suggestions for Authors

The manuscript is sufficiently amended to be published.

Comments on the Quality of English Language

It's generally correct.  Perhaps the fluency of the language could be improved.